# Assessment of Camouflage Effectiveness Based on Perceived Color Difference and Gradient Magnitude

**DOI:** 10.3390/s20174672

**Published:** 2020-08-19

**Authors:** Xueqiong Bai, Ningfang Liao, Wenmin Wu

**Affiliations:** National Laboratory of Colour Science and Engineering, School of Optics and Photonics, Beijing Institute of Technology, Beijing 100081, China; amybx2006@sina.com (X.B.); wu_wm@bit.edu.cn (W.W.)

**Keywords:** image processing, color appearance, camouflage effectiveness

## Abstract

We propose a new model to assess the effectiveness of camouflage in terms of perceived color difference and gradient magnitude. The “image color similarity index” (ICSI) and gradient magnitude similarity deviation (GMSD) were employed to analyze color and texture differences, respectively, between background and camouflage images. Information entropy theory was used to calculate weights for each metric, yielding an overall camouflage effectiveness metric. During the analysis process, both spatial and color perceptions of the human visual system (HVS) were considered, to mimic real-world observations. Subjective tests were used to compare our proposed method with previous methods, and our results confirmed the validity of assessing camouflage effectiveness based on perceived color difference and gradient magnitude.

## 1. Introduction

Camouflage, which serves to blend objects into the background by using similar colors and patterns, has many applications in the fields of bionics and robotics, and for military purposes. Over the past few decades, elements of computer vision, statistical analysis, image processing, nanomaterials, human visual perception, and ergonomics have been introduced to camouflage research [1,2,3,4,5]. A good evaluation method to test the effectiveness of camouflage is very important—one which can provide an effective theoretical basis for camouflage research, predict the performance of camouflage in advance, and help to subsequently optimize the design of camouflage patterns.

Vision-based object detection techniques have been used to discriminate between objects and backgrounds, and thus can also be used to evaluate the effectiveness of camouflage [6,7,8]. These methods, which include the scale invariant feature transform (SIFT) [9], histogram of oriented gradient (HOG) [10], and local binary pattern (LBP) [11], have applications in a variety of fields, such as facial recognition, depth calculation, and three-dimensional reconstruction. When applying these detection methods, it is necessary to consider the characteristics of the human visual system (HVS), to ensure that camouflage is effective for humans. Analyses of the HVS and traditional image quality assessment algorithms have been combined, and the universal image quality index (UIQI) [12] and structural similarity index (SSIM) [13] could be used to obtain results that accord with actual human visual perception. The UIQI and SSIM have been applied to evaluations of camouflage effectiveness based on the similarity between the camouflage and the background. Additional metrics, such as gradient orientation, luminance, and phase congruency, have also been used [14]. However, the computation costs for such metrics can be very high, and they yield only small performance improvements. Gradient magnitude similarity deviation (GMSD) [14] decreases the computational costs and allows for accurate modeling of the HVS. GMSD effectively captures diverse local structures and describes texture patterns in detail. This method has already been applied to image processing and computer vision; however, to the best of our knowledge, it has not been applied to the study of camouflage. In background matching tasks, as well as pattern analysis, color similarities are commonly analyzed. Color matching is critical for survival for many animals in the wild. The methods described above are all based on the differences between the textures of the object and background, and use gray or red, green, and blue (RGB) images. However, RGB color models do not represent the colors actually observed by humans, unless a color appearance model and color management system are used [15].

CIE color space was introduced as a standard for application to displays, art design, color printing, etc. CIELUV and CIELAB, based on CIE color space, aim to achieve a uniform color space with consideration of differences in the perception of colors by the human eye. Perceptual differences between colors are indicated by the distance between points corresponding to individual colors in the color space; these differences can be used as references when assessing the similarity between an object and its background. It has been shown that color perception, in terms of lightness, chroma, and hue, is influenced by surrounding textures and colors. CIE recommends using CIELAB space to characterize colored surfaces and dyes. As the demand for accurate quantification of color differences has increased, more CIELAB color difference formulas have been developed, such as CMC (l:c) [16], BFD(l:c) [17], and CIEDE2000 [18]. CMC (l:c) and CIEDE2000 have been compared, and the camouflage similarity index (CSI) can be used to measure pixel-to-pixel color differences between a camouflaged object and its background using the CIELAB color system [19]. CSI performs well when used to discriminate an object from a large uniform background [20]. To process complex images, Spatial-CIELAB (S-CIELAB) [21] was developed from CIELAB space and takes into account sensitivity to color patterns; it may be superior for analyzing the similarity between a target object and background in assessments of camouflage effectiveness. Color and texture are both important in camouflage design. Therefore, a model that comprehensively evaluates both parameters will provide better results than one that only considers a single parameter.

In this paper, we present a new method for evaluating the effectiveness of camouflage, based on the analysis of color and textural differences between camouflaged objects and their backgrounds. For the color difference analysis, camouflaged objects and backgrounds are transformed into S-CIELAB space. Both spatial and color perceptions of the HVS are considered as the method is based on the way in which the human eye observes the real world. GMSD is applied to assess texture, as it has the advantage of a simple calculation process and yields results that can be related to actual human perception. We designed an experimental procedure using subjective tests to compare performance between the new method and previous ones.

## 2. Methods

### 2.1. Overview

Figure 1 shows our proposed camouflage effectiveness assessment method. Several typical background scenes (N) and camouflage images (M) are input into the model. Both color and texture are analyzed using the “image color similarity index” (ICSI) and GMSD. To calculate the ICSI, all the background and camouflage images are transformed from RGB to S-CIELAB space, and the color difference between each camouflage and background image is obtained based on the standard deviation of the distance in S-CIELAB space. Regarding GMSD, the gradient magnitudes of pairs of images are calculated, and the texture difference is expressed as the standard deviation of the difference in magnitude between the gradients of the camouflage and background images. During the calculation of the ICSI and GMSD, the resolutions of the camouflage and background images are equalized to facilitate comparison. As the ICSI and GMSD are calculated for image pairs, M × N color and texture difference matrices are obtained separately. The weights of each metric are determined using the information entropy method, and the final assessment of the effectiveness of each camouflage pattern is given by the weighted mean of the difference between the camouflage image and all background images. The algorithms of the ICSI, GMSD, and the information entropy method will be described in detail from Section 2.2, Section 2.3 and Section 2.4, respectively.

### 2.2. Image Color Simiarity Index (ICSI)

As mentioned above, S-CIELAB space considers both spatial and color perceptions of the HVS, which enables a more accurate color similarity analysis than other color spaces [21]. Before preprocessing image pairs using S-CIELAB, RGB input images should be transformed into device-independent CIE XYZ tristimulus values [22]. Then, both the camouflage and background image should be converted into the opponent color space. As the HSV can be treated as a low-pass filter from the perspective of imaging processing, low-pass filters for each color channel have been studied in detail in previous works [23]. After spatial filtering, the filtered images are transformed back into CIELAB space, which includes color channels, where *L** represents lightness, *a** is the distance from green to red and *b** is the distance from blue to yellow.

The ICS, which is given by the distance in S-CIELAB color space between the *i*-th camouflaged object and the object-overlapping region of *j*-th background image, can be expressed for pixel (*u*, *v*) as follows.
(1)$$ICS\left(u,v\right)=\sqrt{{\left(L^{*}{}_{i}-L^{*}{}_{j}\right)}^{2}+{\left(a^{*}{}_{i}-a^{*}{}_{j}\right)}^{2}+{\left(b^{*}{}_{i}-b^{*}{}_{j}\right)}^{2}}$$

The standard deviation of the color difference was calculated, as this provides a more comprehensive evaluation than the mean.
(2)$$ICSI=\sqrt{\frac{1}{U\times V}\sum_{u=1}^{U}\sum_{v=1}^{V}{\left(ICS\left(u,v\right)-\frac{1}{U\times V}\sum_{u=1}^{U}\sum_{v=1}^{V}ICS\left(u,v\right)\right)}^{2}}$$

### 2.3. Gradient Magnitude Similarity Deviation (GMSD)

GMSD can be used for texture analysis [14]; we used the Sobel operator for this purpose. The GMSD of the *i*-th camouflage and *j*-th background image can be obtained in the horizontal and vertical directions, as shown in Equations (3)–(5):(3)$$S_{i}\left(u,v\right)=\sqrt{{\left(s_{h}⊗I_{i}\left(u,v\right)\right)}^{2}+{\left(s_{v}⊗I_{i}\left(u,v\right)\right)}^{2}}$$
(4)$$S_{j}\left(u,v\right)=\sqrt{{\left(s_{h}⊗I_{j}\left(u,v\right)\right)}^{2}+{\left(s_{v}⊗I_{j}\left(u,v\right)\right)}^{2}}$$
(5)$$s_{h}=\left[\begin{array}{ccc} -1 & 0 & 1\\ -2 & 0 & 2\\ -1 & 0 & 1 \end{array}\right],\;\;\;\;\;s_{v}=\left[\begin{array}{ccc} -1 & -2 & -1\\ 0 & 0 & 0\\ 1 & 2 & 1 \end{array}\right]$$
where “⊗” denotes the convolution operation.
(6)$$GMS\left(u,v\right)=\left(2S_{i}\left(u,v\right)•S_{j}\left(u,v\right)\right)/\left(S^{2}{}_{i}\left(u,v\right)•S^{2}{}_{j}\left(u,v\right)\right)$$

Standard deviations were summed to compute the overall GMSD, as an index of the textural similarity between the camouflage and background images.
(7)$$GMSD_{ij}=\sqrt{\frac{1}{UV}\sum_{u=1}^{u=U}\sum_{v=1}^{v=V}\left(GMS\left(u,v\right)-\frac{\sum_{u=1}^{u=U}\sum_{v=1}^{v=V}GMS\left(u,v\right)}{UV}\right)}$$

### 2.4. Calculation of Metrics Weights

Information entropy theory can be applied to quantify the amount of useful information contained within data. When the difference between the value of an indicator is high, the entropy is small. Weights are higher for parameters providing more information. As mentioned above, *M* × *N* color and texture difference matrices can be obtained using the ICSI and GMSD methods, respectively.

Using the ICSI and GMSD methods, the difference between the *i*-th camouflage and *j*-th background image is given by *x_ijk_*, where *k* = 1 and 2 for the ICSI and GMSD, respectively. Thus, the proportion of the *i*-th camouflage pattern and *j*-th background pattern calculated using the *k*-th method can be expressed as follows:(8)$$p_{ijk}=x_{ijk}/\sum_{i=1}^{M}\sum_{j=1}^{N}x_{ijk}$$

According to the entropy theory [24], the weight of each metric can be calculated using Equation (9), where the information entropy is defined as *p_ijk_ln*(*p_ijk_*).
(9)$$\omega_{k}=\frac{\left(1+\frac{1}{ln\left(M\times N\right)}\sum_{i=1}^{M}\sum_{j=1}^{N}p_{ijk}ln\left(p_{ijk}\right)\right)}{\sum_{k=1}^{2}\left(1+\frac{1}{ln\left(M\times N\right)}\sum_{i=1}^{M}\sum_{j=1}^{N}p_{ijk}ln\left(p_{ijk}\right)\right)}$$

Using the proposed model, the final camouflage effectiveness value is given by calculating the mean of the M values for color and texture using Equation (10).
(10)$$v_{j}=\sum_{k=1}^{2}\sum_{i=1}^{M}\omega_{k}x_{ijk}$$

## 3. Experimental Setup

To validate our proposed method and compare it to previous ones, we carried out an experiment using an ocean background and several common camouflage patterns. An effectiveness metric for each camouflage pattern was calculated using CSI, UIQI, and our method. To compare the performance of the methods, subjective tests were also carried out comparing the results to HVS assessments.

Figure 2 shows a schematic diagram of the experimental procedure used to validate the developed method. Ten subjects with normal color vision (according to the Ishihara 24-plate test), aged from 22 to 35 years, participated in the experiment. During the experimental process, the participants sat in front of a liquid crystal display (LCD); the set-up was adjusted to ensure a horizontal viewing angle. The distance between the user’s eyes and the screen was set to 55 cm. The background and camouflage images were displayed on the screen. The resolution of the images was 1920 × 1080 (the same as the screen resolution), and clipped camouflage images with a resolution of 300 × 100 were placed within the background image, as shown in Figure 2a. The background images were three images of the sea (under calm, “sparkling”, and rough conditions). Six common camouflage patterns were used, including US woodland, MultiCam, and MARPAT, which have also been used in previous studies. The background and camouflage images that we used are shown in the Appendix A in detail. The background images of the sea were used due to their complex patterns, although other camouflage patterns and background scenes could also be used in conjunction with our method.

During the evaluation process, a clipped camouflage image was placed randomly within a background scene. Each participant was required to locate the camouflage pattern. The “hit rate” and detection time were the performance indicators; task difficulty ratings were also obtained. The hit rate was defined as the percentage of trials on which the camouflaged target was correctly identified (indicated by clicking on it with the computer mouse), and the detection time was the time between the participant clicking the start button to begin the trial and subsequently clicking on the target (or quitting the trial). The task difficulty was rated on a seven-point rating scale at the end of each trial, ranging from 1 (easiest) to 7 (most difficult) [20,25]. Before the experiment, the experimental procedure was explained to all participants. The order of the presentation of stimuli was randomized to avoid order effects. When the participant hit the target or pressed the space bar to quit the trial, the task difficulty rating question appeared on the screen. The experiment had no time limit. There was a 5 min rest period in the middle of the trial to reduce the possibility of visual fatigue.

## 4. Results and Discussion

The Pearson correlation coefficient (PCC) was used to assess the correlations between the results of the objective methods (ICSI, GMSD, and the proposed model), and those of the CSI, UIQI, and the subjective HVS evaluation. Table 1 shows the PCCs of the different methods for hit rate, detection time, and task difficulty. To facilitate the data analysis, we used the absolute PCC values (ranging from 0 to 1), where higher values indicate stronger correlations and more accurate evaluation of camouflage effectiveness.

The PCCs for all three performance parameters (hit rate, detection time, and task difficulty) were higher for our proposed method than for the other methods. Hence, our method performed best in terms of evaluating camouflage effectiveness. This is confirmed by Figure 3, Figure 4 and Figure 5, wherein “x” represents the experimental data for hit rate, detection time, and task difficulty, respectively. In the figures, the proposed method (ICSI + GMSD) shows more linear trends than the other methods. The experimental data for our method are close to the predicted data (blue lines in the figures), and mostly fall within the confidence intervals (the area between the upper and lower curves). Thus, the PCC results showed that the camouflage effectiveness assessment method proposed in this paper was the best-performing method, with results that were consistent with actual human visual perception.

## 5. Conclusions

In this paper, we presented a method for assessing camouflage effectiveness based on perceived color difference and gradient magnitude metrics. Color and texture differences were analyzed while taking both spatial and color perceptions of the HVS into consideration. Color and texture differences were discussed in detail, along with weighting calculations. The PCC data confirmed the superiority of our method over existing ones. In future research, we will apply the proposed method to real-world environments with natural lighting and extend it to encompass the infrared spectrum.

## Figures and Tables

**Figure 1 sensors-20-04672-f001:**
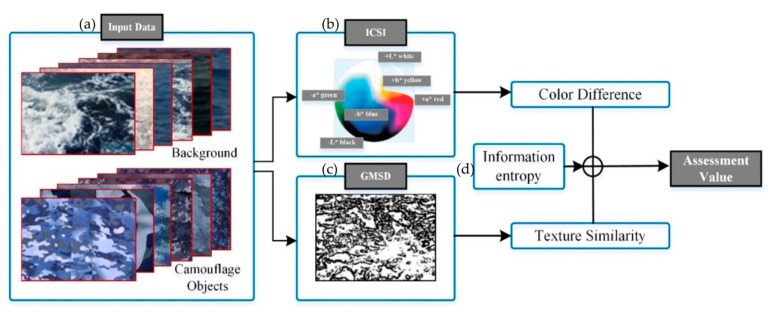
Schematic diagram of the proposed model for assessing camouflage effectiveness: (**a**) Input data of the camouflage object image and the objective-overlapping region of the background image to the model; (**b**) the algorithm of image color similarity index (ICSI); (**c**) The algorithm of gradient magnitude similarity deviation (GMSD); (**d**) Determine the weights of each metric using the information entropy method.

**Figure 2 sensors-20-04672-f002:**
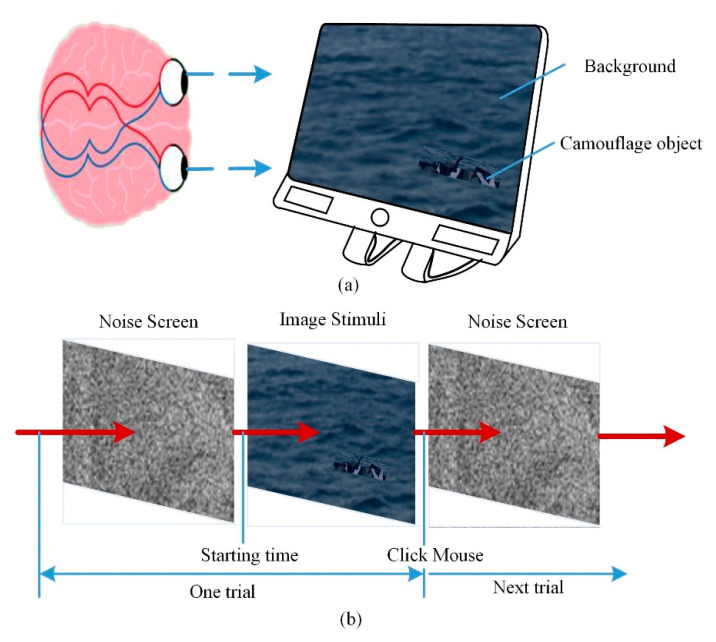
Schematic of the experimental procedure used to validate the developed method: (**a**) the camouflaged image showed in the background on the screen; (**b**) the workflow of each trail.

**Figure 3 sensors-20-04672-f003:**
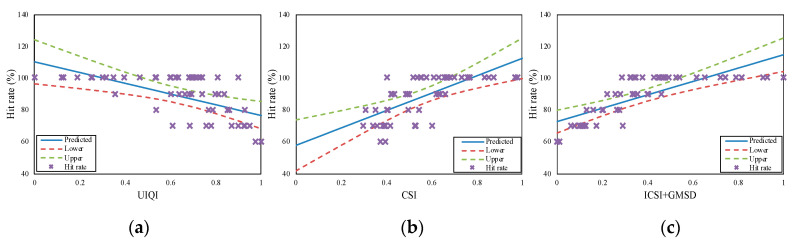
The correlations of the (**a**) camouflage similarity index (CSI), (**b**) universal image quality index (UIQI) and (**c**) image color similarity index and gradient magnitude similarity deviation (ICSI + GMSD) for hit rate.

**Figure 4 sensors-20-04672-f004:**
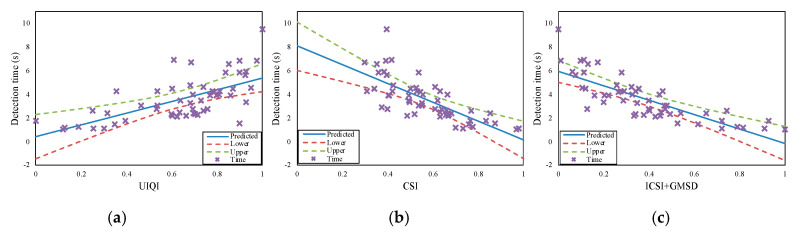
The correlations of the (**a**) CSI, (**b**) UIQI and (**c**) ICSI + GMSD for detection time.

**Figure 5 sensors-20-04672-f005:**
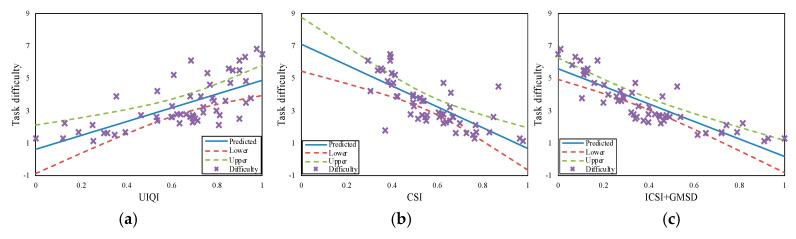
The correlations of the (**a**) CSI, (**b**) UIQI and (**c**) ICSI + GMSD for task difficulty.

**Table 1 sensors-20-04672-t001:** Pearson correlation coefficients of the camouflage effectiveness evaluation methods for various performance parameters.

Index	Hit Rate (%)	Detection Time (Second)	Task Difficulty
UIQI	0.6240	0.6578	0.6882
CSI	0.7586	0.8028	0.8029
GMSD	0.6991	0.7498	0.7952
ICSI	0.6807	0.6887	0.7803
ICSI + GMSD	0.7821	0.8087	0.8637

A *p*-value < 0.05 was taken to indicate statistical significance.

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
