# Peer review of "Assessment of Camouflage Effectiveness Based on Perceived Color Difference and Gradient Magnitude"

_sensors, 2020, doi:10.3390/s20174672_

Round 1
Reviewer 1 Report
In this manuscript, authors propose a new model to assess the effectiveness of camouflage in terms of perceived color difference and gradient magnitude.
The paper can be published in this form if the authors will made some corrections:
- In the Abstract, modify “In this paper, We propose…” with “In this paper, we propose”
- Write the references by the same format.
Author Response
Dear Reviewer,
Thank you for reviewing our work and for providing us with very helpful comments. These comments are helpful for us to revise and improve our paper. We have made a correction in the manuscript according to the reviewer’s comments. All changes are clearly highlighted using the "Track Changes" function in Microsoft Word in the revised manuscript.
Comment 1:
In the Abstract, modify “In this paper, We propose…” with “In this paper, we propose”
Response: We are sorry for our incorrect writing, and we have corrected it in the revised manuscript.
Comment 2:
Write the references in the same format.
Response: Thank you for pointing this out. The format of the references has been carefully corrected in the revised manuscript.
Thank you once again for all your help.
Best wishes,
Xueqiong Bai, on behalf of all the co-authors
Reviewer 2 Report
This manuscript proposes a method for assessing the camouflage effectiveness using the ICSI and GMSD metrics. The work is interesting. This reviewer have the following concerns:
- It seems that Eq. 1 computes pixel-wise ICS values. As the camouflage object is much smaller than the background, the computation is applied between the object and the object-overlapping region of the background? This should be clarified since it determines if the color difference computation is reasonable.
- 2 indicates that the camouflage object image is directly inserted into the background. I think more careful treatment is necessary in the experiments in order to make the simulation more close real scenario. First, the surrounding region of the object should be removed such that the object is seamlessly inserted into the ocean background. Second, after inserting the object, the contour region between the object and the background should be further blurred to simulating a distant observation.
- How many backgrounds (ocean or others) have been used in the experiments? The authors should illustrates more backgrounds and camouflage objects (before and after fusion) such that the readers can understand the difficult level of object discrimination. This is actually directly related to the hit rate reported in the manuscript.
- The authors should analyze which metric is more important in the assessment, ICSI or GMSD?
Author Response
Dear Reviewer,
Thank you very much for your careful review concerning our manuscript “Assessment of camouflage effectiveness based on perceived color difference and gradient magnitude” (ID: sensors-858728). These comments are all valuable and helpful for revising and improving our paper, as well as the importance of guiding significance to our researches. We have studied these comments carefully and have made corrections which we hope meet with approval. All changes are clearly highlighted using the "Track Changes" function in Microsoft Word in the revised manuscript. The response to the reviewer’s comments is uploaded as a PDF file. Please see the attachment.
We appreciate Reviewer's warm work earnestly and hope the correction will meet with approval.
Best wishes,
Xueqiong Bai, on behalf of all the co-authors

Reviewer 3 Report
The authors reported a new model to assess the effectiveness of camouflage in terms of perceived color difference and gradient magnitude. The “image color similarity index” (ICSI) and gradient magnitude similarity deviation (GMSD) were employed to analyze color and texture differences, respectively, between background and camouflage images. The observations are as follow. 1. Re-write the abstract in order to be more conscious, the abstract contains a part of background section that must be placed in the introduction, ie: “ Studies on camouflage designs have found many applications in bionics, robotics, and military. Camouflage effectiveness assessment methods plays an important role, which can guide and optimize the design. Vision-based object detection techniques have been utilized in detecting object from background, which can also evaluate the camouflage effectiveness. Existing methods cannot obtain the actual color as human observes.” The abstract should reflect background, objectives, methodology and results in a most summarized form to give readers a complete view of the work presented, in addition the conclusions. 2. In the introduction section , line 31, what is the meaning for the sentence: “Good methods to assess camouflage effectiveness and optimize designs are important”. 3. In figure 1, describe in detail each block for the model. 4. In line 95, the used entropy method must be described. 5. Explain the reason to use 10 subjects. 6. Some evidence in supplementary material must be included about the experimental set-up, such as videos, etc etc
Author Response
Dear Reviewer,
Thank you very much for your careful review and constructive suggestion with regard to our manuscript “Assessment of camouflage effectiveness based on perceived color difference and gradient magnitude” (ID: sensors-858728). These comments are all valuable and helpful for revising and improving our paper, as well as the importance of guiding significance to our researches. We have studied these comments carefully and have made corrections which we hope meet with approval. All changes are clearly highlighted using the "Track Changes" function in Microsoft Word in the revised manuscript. The response to the reviewer’s comments is uploaded as a PDF file. Please see the attachment.
We appreciate Reviewer's warm work earnestly and hope the correction will meet with approval.
Best wishes,
Xueqiong Bai, on behalf of all the co-authors

Round 2
Reviewer 2 Report
The authors have revised the ms according to my suggestions.